# A simple normative network approximates local non-Hebbian learning in the cortex

**Siavash Golkar** [1]     **David Lipshutz** [1]     **Yanis Bahroun** [1]

**Anirvan M. Sengupta** [1,2]     **Dmitri B. Chklovskii** [1,3]

[1] Center for Computational Neuroscience, Flatiron Institute
[2] Department of Physics and Astronomy, Rutgers University
[3] Neuroscience Institute, NYU Medical Center

{sgolkar,dlipshutz,ybahroun,mitya}@flatironinstitute.org
anirvans.physics@gmail.com

## Abstract

To guide behavior, the brain extracts relevant features from high-dimensional data streamed by sensory organs. Neuroscience experiments demonstrate that the processing of sensory inputs by cortical neurons is modulated by instructive signals which provide context and task-relevant information. Here, adopting a normative approach, we model these instructive signals as supervisory inputs guiding the projection of the feedforward data. Mathematically, we start with a family of Reduced-Rank Regression (RRR) objective functions which include Reduced Rank (minimum) Mean Square Error (RRMSE) and Canonical Correlation Analysis (CCA), and derive novel offline and online optimization algorithms, which we call Bio-RRR. The online algorithms can be implemented by neural networks whose synaptic learning rules resemble calcium plateau potential dependent plasticity observed in the cortex. We detail how, in our model, the calcium plateau potential can be interpreted as a backpropagating error signal. We demonstrate that, despite relying exclusively on biologically plausible local learning rules, our algorithms perform competitively with existing implementations of RRMSE and CCA.

## 1   Introduction

In the brain, extraction of behaviorally-relevant features from high-dimensional data streamed by sensory organs occurs in multiple stages. Early stages of sensory processing, e.g., the retina, lack feedback and are naturally modeled by unsupervised learning algorithms [1]. In contrast, subsequent processing by cortical circuits is modulated by instructive signals from other cortical areas [2], which provide context and task-related information [3], thus calling for supervised learning models.

Unsupervised models of early sensory processing, despite employing many simplifying assumptions, have successfully bridged the salient features of biological neural networks, such as the architecture, synaptic learning rules and receptive field structure, with computational tasks such as dimensionality reduction, decorrelation, and whitening [4, 5, 6, 7, 8]. The success of such models was driven by two major factors. First, following a normative framework, their synaptic learning rules, network architecture and activity dynamics were derived by optimizing a principled objective, leading to an analytic understanding of the circuit computation without the need for numerical simulation [9]. Second, these models went beyond purely theoretical explorations by appealing to and explaining various experimental observations of early sensory organs available at the time [5, 8, 9].

In contrast to early sensory processing, subsequent processing in the cortex (both neocortex [10, 2, 11, 12, 13] and hippocampus [14, 15, 16, 17]) is guided by supervisory signals. In particular, in cortical pyramidal neurons, proximal dendrites receive and integrate feedforward inputs leading to the generation of action potentials (i.e., the output of the neuron). The distal dendrites of the apical tuft, in contrast, receive and integrate instructive signals resulting in local depolarization. When the local depolarization is large relative to inhibitory currents, this generates a calcium plateau potential that propagates throughout the entire neuron. If the calcium plateau coincides with feedforward input, it strengthens corresponding proximal synapses, thereby providing an instructive signal in these circuits [12, 11, 15, 16].

In this work, we model cortical processing as a projection of feedforward sensory input that is modulated by instructive signals from other cortical areas. Inspired by the success of the normative approach in early sensory processing, we adopt it here. Mathematically, the projections of sensory input can be learned by minimizing the prediction error or maximizing the correlation of the projected input with the instructive signal. These correspond to two instances of the Reduced-Rank Regression (RRR) objectives: Reduced-Rank (minimum) Mean Square Error (RRMSE) [18] and Canonical Correlation Analysis (CCA) [19].

To serve as a viable model of brain function, an algorithm must satisfy at least the following two criteria [9]. First, because sensory inputs are streamed to the brain and require real-time processing, it must be modeled by an online learning algorithm that does not store any significant fraction of the data. To satisfy this requirement, unlike standard offline formulations, which output projection matrices, at each time step, the algorithm must compute the projection from the input of that time step. The projection matrices are updated at each time step and can be represented in synaptic weights. Second, a neural network implementation of such an algorithm must rely exclusively on local synaptic learning rules. Here, locality means that the plasticity rules depend exclusively on the variables available to the biological synapse, i.e., the physicochemical activities of the pre- and post-synaptic neurons in the synaptic neighborhood. The Hebbian update rule is an example of local learning, where the change of synaptic weight is proportional to the correlation between the output activities of the pre- and post-synaptic neurons [20].

**Contributions**

- We derive novel algorithms for a family of RRR problems, which include RRMSE and CCA, and implement them in biologically plausible neural networks that resemble cortical micro-circuits.
- We demonstrate within the confines of our model how the calcium plateau potential in cortical microcircuits encodes a backpropagating error signal.
- We show numerically on a real-world dataset that our algorithms perform competitively compared with current state-of-the-art algorithms.

## 2 Related works

Our contributions are related to several lines of computational and theoretical research. One of the earliest normative models of cortical computation is based on the predictive coding framework where the feedback attempts to predict the feedforward input. When trained on natural images, this approach can explain extra-classical response properties observed in the visual cortex [21, 22]. The predictive coding framework has recently been used for the supervised training of deep networks with Hebbian learning rules [23]. However, these models have not been mapped onto the anatomy and physiology, especially the non-Hebbian synaptic plasticity, of cortical microcircuits [15, 16].

A prescient paper [24] proposed that supervised learning in the cortex can be implemented by multi-compartmental pyramidal neurons with non-Hebbian learning rules driven by calcium plateau potentials. Building on this proposal, [25, 26, 27] demonstrated possible biological implementations of backpropagation in deep networks. Neuroscience experiments have motivated the development of several biologically realistic models of microcircuits with multi-compartmental neurons and non-Hebbian learning rules [28, 29, 30]. Specifically, [29, 30] showed that calcium plateau potentials, generated in the apical tuft, can modulate the efficacy of proximal synapses. These demonstrations, however, are limited in that they were shown analytically in a small region of parameter space or they rely entirely on numerical simulations.

In the context of statistical learning, multiple RRMSE [31, 32, 33, 34, 35] and CCA [36, 37, 38, 39, 40, 41] algorithms have been developed. Of these algorithms, none satisfy the minimal criteria for biological plausibility. Biologically plausible formulations of CCA, as an unsupervised data integration algorithm following the normative approach, were proposed using deflation [42] and fully online in [43].

## 3 An objective function for reduced-rank regression problems

In this section, we review the Reduced-Rank Regression (RRR) problem which encompasses Canonical Correlation Analysis (CCA) and Reduced Rank (minimum) Mean Square Error (RRMSE) as special cases.

**Notation.** For positive integers $m, n$, let $\mathbb{R}^m$ denote $m$-dimensional Euclidean space, and let $\mathbb{R}^{m \times n}$ denote the set of $m \times n$ real-valued matrices. We use boldface lower-case letters (e.g., $\mathbf{v}$) to denote vectors and boldface upper-case letters (e.g., $\mathbf{M}$) to denote matrices. Let $\mathbf{I}_m$ denote the $m \times m$ identity matrix.

Let $\{(\mathbf{x}_t, \mathbf{y}_t)\}_{t=1}^T$ be a sequence of pairs of data points with $\mathbf{x}_t \in \mathbb{R}^m$, $\mathbf{y}_t \in \mathbb{R}^n$. We refer to $\mathbf{x}_t$ as the predictor variable and $\mathbf{y}_t$ as the response variable. Define the data matrices $\mathbf{X} := [\mathbf{x}_1, \dots, \mathbf{x}_T] \in \mathbb{R}^{m \times T}$ and $\mathbf{Y} := [\mathbf{y}_1, \dots, \mathbf{y}_T] \in \mathbb{R}^{n \times T}$. Let $\mathbf{C}_{xx} := \frac{1}{T} \mathbf{X} \mathbf{X}^\top$, $\mathbf{C}_{yy} := \frac{1}{T} \mathbf{Y} \mathbf{Y}^\top$, and $\mathbf{C}_{xy} := \frac{1}{T} \mathbf{X} \mathbf{Y}^\top$ be the empirical covariance matrices. Throughout this paper, we assume that $\mathbf{X}$ and $\mathbf{Y}$ are centered and full rank.

### 3.1 Problem formulation

The goal of RRR is to find a low-rank projection matrix $\mathbf{P} \in \mathbb{R}^{n \times m}$ that minimizes the error between $\mathbf{PX}$ and $\mathbf{Y}$. The low-rank constraint favors the extraction of features that are most predictive of the response variables, thus preventing over-fitting [18]. We can formalize this as follows:

$$\underset{\mathbf{P} \in \mathbb{R}^{n \times m}}{\arg \min} \frac{1}{T} \big\| \mathbf{Y} - \mathbf{PX} \big\|_\Sigma^2 \quad \text{subject to} \quad \text{rank}(\mathbf{P}) \le k, \tag{1}$$

where $k \le \min(m, n)$ determines the rank of the problem, $\Sigma \in \mathbb{R}^{n \times n}$ is a positive definite matrix, and $\| \cdot \|_\Sigma$ is the $\Sigma$-norm defined by $\|\mathbf{A}\|_\Sigma^2 := \text{Tr } \mathbf{A}^\top \Sigma \mathbf{A}$ for $\mathbf{A} \in \mathbb{R}^{n \times T}$. Intuitively, the $\Sigma$-norm is a generalized norm that can take into account the noise statistics of the samples [44]. Two common choices for $\Sigma$ are $\Sigma = \mathbf{I}_n$ and $\Sigma = \mathbf{C}_{yy}^{-1}$. When $\Sigma = \mathbf{I}_n$, the RRR problem reduces to minimizing the mean square error (MSE) with a low-rank constraint. We refer to this objective as Reduced Rank (minimum) Mean Square Error (RRMSE) [18].[1] For $\Sigma = \mathbf{C}_{yy}^{-1}$, the objective in Eq. (1) is equivalent to Canonical Correlation Analysis (CCA) (see Sec. A of the supplementary materials).

### 3.2 Parametrizing the projection matrix

The low-rank constraint, $\text{rank}(\mathbf{P}) \le k$, in Eq. (1) can be enforced by expressing $\mathbf{P} = \Sigma^{-1} \mathbf{V}_y \mathbf{V}_x^\top$, where $\mathbf{V}_x \in \mathbb{R}^{m \times k}$ and $\mathbf{V}_y \in \mathbb{R}^{n \times k}$ (the inclusion of $\Sigma^{-1}$ here is for convenience in the derivation below). The matrix $\mathbf{V}_x^\top$ projects the inputs $\mathbf{x}_t$ onto a $k$-dimensional subspace and the column vectors of $\Sigma^{-1} \mathbf{V}_y$ span the range of the projection matrix $\mathbf{P}$. Plugging into Eq. (1), we have

$$\min_{\mathbf{V}_x \in \mathbb{R}^{m \times k}} \min_{\mathbf{V}_y \in \mathbb{R}^{n \times k}} \frac{1}{T} \big\| \mathbf{Y} - \Sigma^{-1} \mathbf{V}_y \mathbf{V}_x^\top \mathbf{X} \big\|_\Sigma^2. \tag{2}$$

The minimum of this objective is not unique: given a solution $(\mathbf{V}_x, \mathbf{V}_y)$ and any invertible matrix $\mathbf{M} \in \mathbb{R}^{k \times k}$, $(\mathbf{V}_x \mathbf{M}^\top, \mathbf{V}_y \mathbf{M}^{-1})$ is also a solution. To constrain the solution set, we impose the whitening constraint $\mathbf{V}_x^\top \mathbf{C}_{xx} \mathbf{V}_x = \mathbf{I}_k$. Expanding the quadratic in (2), dropping terms that do not depend on $\mathbf{V}_x$ or $\mathbf{V}_y$, and using the whitening constraint, we arrive at

$$\min_{\mathbf{V}_x \in \mathbb{R}^{m \times k}} \min_{\mathbf{V}_y \in \mathbb{R}^{n \times k}} \text{Tr}(\mathbf{V}_y^\top \Sigma^{-1} \mathbf{V}_y - 2 \mathbf{V}_x^\top \mathbf{C}_{xy} \mathbf{V}_y) \quad \text{subject to} \quad \mathbf{V}_x^\top \mathbf{C}_{xx} \mathbf{V}_x = \mathbf{I}_k. \tag{3}$$

The output of our algorithms will be the low-rank projection of $\mathbf{X}$, which we call $\mathbf{Z} := \mathbf{V}_x^\top \mathbf{X}$. Intuitively, for RRMSE ($\boldsymbol{\Sigma} = \mathbf{I}_n$), optimization of this objective would find $\mathbf{Z}$ which is most informative, in terms of MSE loss, of the response variable $\mathbf{Y}$. For CCA ($\boldsymbol{\Sigma} = \mathbf{C}_{yy}^{-1}$), optimization of this objective finds the projection $\mathbf{Z}$ which has the highest correlation with the response variable $\mathbf{Y}$.

We parametrize the normalizing matrix by its inverse as $\boldsymbol{\Sigma}^{-1} = \boldsymbol{\Sigma}_s^{-1} := s\,\mathbf{C}_{yy} + (1-s)\,\mathbf{I}_n$ with $0 \leq s \leq 1$. RRR with this normalizing matrix corresponds to a family of objectives which interpolate between RRMSE at $s = 0$ and CCA at $s = 1$.

## 4 Algorithm derivation

In this section, starting from Eq. (3), we derive offline and online algorithms for the family of RRR objectives parametrized by $s$.

### 4.1 Offline algorithms

Noting that imposing the constraint $\mathbf{V}_x^\top \mathbf{C}_{xx} \mathbf{V}_x = \mathbf{I}_k$ via a Lagrange multiplier leads to non-local update rules (see Sec. B of the supplementary materials), following [45] we impose the weaker inequality constraint $\mathbf{V}_x^\top \mathbf{C}_{xx} \mathbf{V}_x \preceq \mathbf{I}_k$ by introducing the matrix $\mathbf{Q} \in \mathbb{R}^{k \times k}$

$$\min_{\mathbf{V}_x \in \mathbb{R}^{m \times k}} \min_{\mathbf{V}_y \in \mathbb{R}^{n \times k}} \max_{\mathbf{Q} \in \mathbb{R}^{k \times k}} \operatorname{Tr} \mathbf{V}_y^\top \boldsymbol{\Sigma}_s^{-1} \mathbf{V}_y - 2\mathbf{V}_x^\top \mathbf{C}_{xy} \mathbf{V}_y + \mathbf{Q}\mathbf{Q}^\top (\mathbf{V}_x^\top \mathbf{C}_{xx} \mathbf{V}_x - \mathbf{I}_k), \quad (4)$$

where $\mathbf{Q}\mathbf{Q}^\top$ is the positive semi-definite Lagrange multiplier enforcing the inequality. As in [45], the dynamics of the optimization enforce that the inequality constraint is saturated, i.e., $\mathbf{V}_x^\top \mathbf{C}_{xx} \mathbf{V}_x = \mathbf{I}_k$ is satisfied at the optimum of the objective (for a different proof see Sec. C). In the offline setting, objective (4) can be optimized using gradient descent-ascent dynamics derived by taking partial derivatives:

$$\mathbf{V}_x^\top \leftarrow \mathbf{V}_x^\top + \eta(\mathbf{V}_y^\top \mathbf{C}_{yx} - \mathbf{Q}\mathbf{Q}^\top \mathbf{V}_x^\top \mathbf{C}_{xx}) \tag{5}$$

$$\mathbf{V}_y^\top \leftarrow \mathbf{V}_y^\top + \eta(\mathbf{V}_x^\top \mathbf{C}_{xy} - \mathbf{V}_y^\top \boldsymbol{\Sigma}_s^{-1}) \tag{6}$$

$$\mathbf{Q} \leftarrow \mathbf{Q} + \frac{\eta}{\tau}(\mathbf{V}_x^\top \mathbf{C}_{xx} \mathbf{V}_x - \mathbf{I}_k)\mathbf{Q}, \tag{7}$$

where $\eta > 0$ is the learning rate for $\mathbf{V}_x$ and $\mathbf{V}_y$, and $\tau > 0$ is a parameter controlling the ratio of the descent and ascent steps.

### 4.2 Online algorithms

In the online (or streaming) setting, the input is presented one sample at a time, and the algorithm must find the projection without storing any significant fraction of the dataset.

To derive an online algorithm, we rewrite the objective function (4) making the dependence of the objective on each individual sample manifest:

$$\min_{\mathbf{V}_x} \min_{\mathbf{V}_y} \max_{\mathbf{Q}} \frac{1}{T} \sum_{t=1}^{T} \mathbf{V}_y^\top (s\mathbf{y}\mathbf{y}^\top + (1-s)\mathbf{I}_n)\mathbf{V}_y - 2\mathbf{V}_x^\top \mathbf{x}_t \mathbf{y}_t^\top \mathbf{V}_y + \mathbf{Q}\mathbf{Q}^\top (\mathbf{V}_x^\top \mathbf{x}_t \mathbf{x}_t^\top \mathbf{V}_x - \mathbf{I}_k). \tag{8}$$

If we now perform stochastic gradient descent/ascent [46], i.e., perform the gradient updates with respect to individual samples, we arrive at our online algorithm. Explicitly, at time $t$, we have:

$$\mathbf{V}_x^\top \leftarrow \mathbf{V}_x^\top + \eta(\mathbf{a}_t - \mathbf{Q}\mathbf{n}_t)\mathbf{x}_t^\top \tag{9}$$

$$\mathbf{V}_y^\top \leftarrow \mathbf{V}_y^\top + \eta(\mathbf{z}_t \mathbf{y}_t^\top - s\,\mathbf{a}_t \mathbf{y}_t^\top - (1-s)\,\mathbf{V}_y^\top) \tag{10}$$

$$\mathbf{Q} \leftarrow \mathbf{Q} + \frac{\eta}{\tau}(\mathbf{z}_t \mathbf{n}_t^\top - \mathbf{Q}). \tag{11}$$

where $\mathbf{z}_t := \mathbf{V}_x^\top \mathbf{x}_t$ is the output of the algorithm, $\mathbf{a}_t := \mathbf{V}_y^\top \mathbf{y}_t$ and $\mathbf{n}_t := \mathbf{Q}^\top \mathbf{z}_t$.

Our algorithms, which we call Bio-RRR, are summarized in Alg. 1 (offline) and Alg. 2 (online).

**Algorithm 1:** Offline Bio-RRR

**input:** $\mathbf{X} \in \mathbb{R}^{m \times T}$, $\mathbf{Y} \in \mathbb{R}^{n \times T}$
**initialize** $\mathbf{V}_x$, $\mathbf{V}_y$, and $\mathbf{Q}$.
$\mathbf{C}_{xx} \leftarrow \mathbf{X}\mathbf{X}^\top / T$ ; $\mathbf{C}_{xy} \leftarrow \mathbf{X}\mathbf{Y}^\top / T$
$\mathbf{\Sigma}_s^{-1} \leftarrow s\,\mathbf{Y}\mathbf{Y}^\top / T + (1-s)\,\mathbf{I}_n$
**repeat:**
$\quad \mathbf{V}_x^\top \leftarrow \mathbf{V}_x^\top + \eta(\mathbf{V}_y^\top \mathbf{C}_{yx} - \mathbf{Q}\mathbf{Q}^\top \mathbf{V}_x^\top \mathbf{C}_{xx})$
$\quad \mathbf{V}_y^\top \leftarrow \mathbf{V}_y^\top + \eta(\mathbf{V}_x^\top \mathbf{C}_{xy} - \mathbf{V}_y^\top \mathbf{\Sigma}_s^{-1})$
$\quad \mathbf{Q} \leftarrow \mathbf{Q} + \frac{\eta}{\tau}(\mathbf{V}_x^\top \mathbf{C}_{xx} \mathbf{V}_x - \mathbf{I}_k)\mathbf{Q}$
**until** convergence
**output:** $\mathbf{Z} = \mathbf{V}_x^\top \mathbf{X}$ $\quad \triangleright$ projected predictor

**Algorithm 2:** Online Bio-RRR

**input:** $\mathbf{x}_t \in \mathbb{R}^m$, $\mathbf{y}_t \in \mathbb{R}^n$ $\quad \triangleright$ new sample
$\qquad\quad \mathbf{V}_x, \mathbf{V}_y, \mathbf{Q}$ $\quad \triangleright$ previous matrices
$\mathbf{z}_t \leftarrow \mathbf{V}_x^\top \mathbf{x}_t$; $\quad \mathbf{n}_t \leftarrow \mathbf{Q}^\top \mathbf{z}_t$; $\quad \mathbf{a}_t \leftarrow \mathbf{V}_y^\top \mathbf{y}_t$
$\mathbf{V}_x^\top \leftarrow \mathbf{V}_x^\top + \eta(\mathbf{a}_t - \mathbf{Q}\mathbf{n}_t)\mathbf{x}_t^\top$
$\mathbf{V}_y^\top \leftarrow \mathbf{V}_y^\top + \eta(\mathbf{z}_t\mathbf{y}^\top - s\,\mathbf{a}_t\mathbf{y}^\top - (1-s)\,\mathbf{V}_y^\top)$
$\mathbf{Q} \leftarrow \mathbf{Q} + \frac{\eta}{\tau}(\mathbf{z}_t\mathbf{n}_t^\top - \mathbf{Q})$
**output:** $\mathbf{z}_t$ $\quad \triangleright$ projected sample
$\qquad\quad \mathbf{V}_x, \mathbf{V}_y, \mathbf{Q}$ $\quad \triangleright$ updated matrices

## 5 Biological implementation and comparison with experiment

In this section, we introduce a biological neural circuit that implements the online RRR algorithm and demonstrate that the details of this circuit resemble neurophysiological properties of pyramidal cells in the neocortex and the hippocampus.

### 5.1 Neural circuit

The algorithm for online RRR summarized by the update rules in Eqs. (9)−(11) can be implemented in a neural circuit with schematic shown in Fig. 1. In this circuit, the individual components of the output of Bio-RRR, $z_1, \ldots, z_k$, are represented as the outputs of $k$ neurons. The matrices $\mathbf{V}_x$ and $\mathbf{V}_y$ are encoded as the weights of synapses between the output neurons and the inputs of the network (blue and pink nodes in Fig. 1). Explicitly the element $V_x^{ij}$ (resp. $V_y^{ij}$) is the efficacy of the synapse connecting $x_i$ (resp. $y_i$) to the $j^{\text{th}}$ output neuron $z_j$. Because of the disjoint nature of the two inputs, we model these as synapsing respectively onto the distal (apical tuft) and proximal (mostly basal) dendrites of the output neurons, Fig. 1 . The quantities $\mathbf{z}_t = \mathbf{V}_x^\top \mathbf{x}_t$ and $\mathbf{a}_t = \mathbf{V}_y^\top \mathbf{y}_t$ are then the integrated dendritic currents in each dendritic compartment.

Similarly, the auxiliary variable $\mathbf{n}$ is represented by the activity of $k$ interneurons with $\mathbf{Q}$ encoded in the weights of synapses connecting $\mathbf{n}$ to $\mathbf{z}$ (purple nodes on the upper dendritic branch of $\mathbf{z}$) and $\mathbf{Q}^\top$ encoded in the weights of synapses from $\mathbf{z}$ to $\mathbf{n}$ (gray nodes). In a biological setting, the implied equality of weights of synapses from $\mathbf{z}$ to $\mathbf{n}$ and the transpose of those from $\mathbf{n}$ to $\mathbf{z}$ can be guaranteed approximately by application of the same Hebbian learning rule (see supplementary materials Sec. D).

The proximal synaptic weights, given by the elements of $\mathbf{V}_x$, are updated by the product of two factors represented in the corresponding post- and pre-synaptic neurons (Eq. 9).

$$\delta\mathbf{V}_x^\top \propto (\mathbf{a}_t - \mathbf{Q}\mathbf{n}_t)\mathbf{x}_t^\top$$

The first factor $(\mathbf{a}_t - \mathbf{Q}\mathbf{n}_t)$, is the difference between the excitatory synaptic current in the apical tuft $(\mathbf{a}_t = \mathbf{V}_y^\top \mathbf{y}_t)$ and the inhibitory current induced by interneurons synapsing onto the distal compartment $(\mathbf{Q}\mathbf{n}_t)$. Biologically, this factor can be approximated by the calcium plateau potential traveling down the apical shaft. The second factor is the input $\mathbf{x}_t$ to the proximal dendrites. Therefore, the synaptic weight update is proportional to quantities that are available to the synapse locally.

The synaptic learning rule for $\mathbf{V}_y$ (Eq. 10) also involves the products of pre- and post-synaptic variables but weighted by the parameter $s$,

$$\delta\mathbf{V}_y^\top \propto \left[ \mathbf{z}_t\mathbf{y}_t^\top - (1-s)\,\mathbf{V}_y^\top - s\,\mathbf{a}_t\mathbf{y}_t^\top \right]$$

In the case of RRMSE ($s = 0$), the update is Hebbian ($\mathbf{z}_t\mathbf{y}_t^\top$) with a homeostasis decay term ($-\mathbf{V}_y^\top$). In the case of CCA ($s = 1$), the synaptic weight update is proportional to $(\mathbf{z}_t - \mathbf{a}_t)\mathbf{y}^\top$, where the difference between the (dendritically backpropagated) output activity of the pyramidal neuron ($\mathbf{z}_t$) and the total synaptic input to the distal compartment ($\mathbf{a}_t$) can be computed in the corresponding

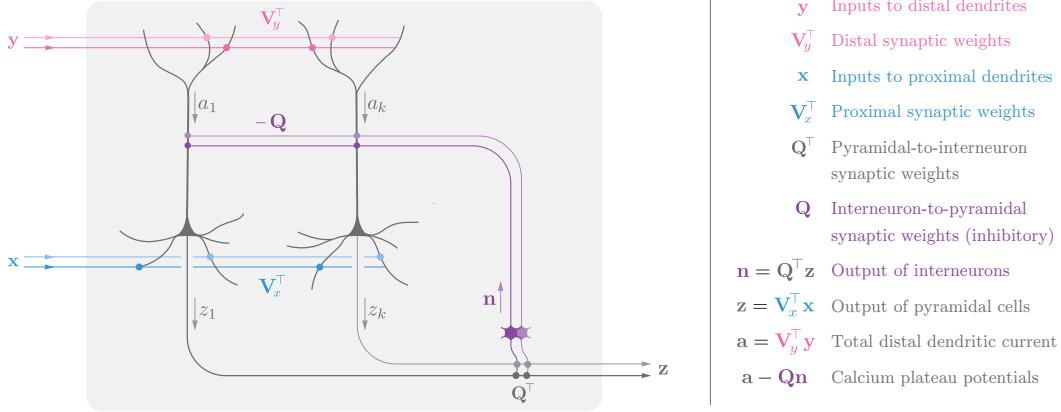

Figure 1: Cortical microcircuit for Bio-RRR. Pyramidal neurons (black) receive inputs $\mathbf{x}$ onto the dendrites proximal to the cell bodies (black triangles) weighted by $\mathbf{V}_x^\top$, and inputs $\mathbf{y}$ onto the distal dendrites weighted by $\mathbf{V}_y^\top$. The calcium plateau potential is the difference between the total distal dendritic current for each pyramidal neuron, $\mathbf{a} = (a_1, \ldots, a_k)$, and the corresponding component of the inhibitory input, $-\mathbf{Qn}$. Output activity of pyramidal neurons, $\mathbf{z} = (z_1, \ldots, z_k)$, is fed back via inhibitory interneurons (purple). The equivalence of the pyramidal-to-interneuron weight matrix, $\mathbf{Q}^\top$, and the transpose of the interneuron-to-pyramidal weight matrix, $\mathbf{Q}$, follows from the operation of the local learning rules, see Sec. D of supplementary materials.

post-synaptic neuron (cf. [28]). In the intermediate cases of $0 < s < 1$, the update rule for $\mathbf{V}_y$ linearly interpolates between these two cases and remains local.

Finally, this circuit has the advantage of being purely feedforward in the sense that the output computation does not require equilibration of recurrent activity in lateral connections as was the case in e.g. [9]. This is due to the segregation between the proximal compartment that computes the output of the neuron and the distal compartment which receives the inhibitory lateral feedback.

## 5.2 Comparison with neuroscience experiments

The Bio-RRR circuit derived above has many features in common with cortical microcircuits but also deviates from them in a number of ways. Microcircuits in the cortex contain two classes of neurons: excitatory pyramidal neurons and inhibitory interneurons.[2] The pyramidal neurons can be considered the output neurons as their axon projections leave the local circuit. Similar to the output neurons of our circuit in Fig. 1, pyramidal neurons have two integration sites, the proximal compartment comprised of the basal and proximal apical dendrites providing inputs to the soma, and the distal compartment comprised of the apical dendritic tuft [49, 3]. These two compartments receive excitatory inputs from two separate sources [50, 2].

The inputs onto the two compartments are processed differently [2, 51, 52, 49, 3]. The proximal inputs directly drive the pyramidal neuron output by generating action potentials. If the distal inputs are stronger than the inhibitory post-synaptic currents driven by the interneurons, they generate a calcium plateau potential, which can also cause action potentials in the pyramidal neurons [2]. This is in contrast to our RRR algorithms, where only the proximal input contributes to the output, $\mathbf{z} = \mathbf{V}_x^\top \mathbf{x}$. Neglecting the contribution of the apical inputs to the action potential generation can be justified by the temporal sparsity of calcium plateau potentials. The situation where both proximal and distal inputs contribute significantly to the generation of action potentials can be modeled by an alternative biologically plausible implementation of CCA [43].

The calcium plateau potentials generated by the apical tuft inputs drive the plasticity of proximal synapses [12, 14, 15, 16]. Because this update is not purely dependent on the action potentials of the pre- and post-synaptic neurons, such plasticity is called non-Hebbian [16]. This resembles the synaptic updates of $\mathbf{V}_x$ in Eq. (9). However, while the teaching signal for the proximal synapses in Bio-RRR (i.e., $\mathbf{a}_t - \mathbf{Qn}_t$) is signed and graded, in the cortex, these signals are generally believed to be stereotypical [2]. Graded calcium mediated signals were recently observed in [29].

Whereas the pyramidal neurons of the cortex fire all-or-nothing action potentials, Bio-RRR neurons are analog and linear as in firing-rate models. Furthermore, the goal of the RRR objectives is to reduce the dimensionality of the feedforward input, whereas sensory cortical processing is thought to expand dimensionality [53, 54, 55, 56]. These two disparities between our networks and realistic circuits are closely linked in that it is impossible to perform meaningful dimensionality expansion with linear neurons. However, due to the analytical tractability of simplified linear models, they provide insights that are difficult to obtain in more realistic models amenable only to numerical simulations.

The above comparisons of our algorithm with experiment apply equally to Bio-RRR with any $0 \leq s \leq 1$. The property which distinguishes different members of this family of algorithms is the update rule associated with the synapses of the distal compartment $\mathbf{V}_y$ given in Eq. (10). There is conflicting experimental evidence regarding the plasticity of the distal apical dendrites in different areas of the brain. In the neocortex, the plasticity is thought to be Hebbian [11, 57], whereas in the hippocampus, experimental evidence points to non-Hebbian plasticity [12]. As discussed in the previous section, our online RRMSE and CCA algorithms require that distal synapses follow Hebbian and non-Hebbian plasticity rules, respectively. For a given cortical circuit, determining whether CCA or RRMSE or some intermediate value of $s$ provides the best fit would require a close examination of the plasticity rules of the distal compartment.

## 6 Interpretation of calcium plateau potential in Bio-RRR

Experimentally, the calcium plateau potentials act as instructive signals in cortical pyramidal neurons by driving plasticity in the proximal dendrites [15, 16, 30]. Several prior works [24, 25, 26, 27] have suggested that the calcium plateau potential carries the backpropagation error. Here, we show that the calcium plateau potential plays a similar role in Bio-RRR provided the network is close to the optimum of the objective. In the process, we will also show how Bio-RRR avoids the weight transport problem of Artificial Neural Networks (ANNs) trained with the backpropagation algorithm (backprop).

We first describe how a two-layer ANN trained with backprop would implement RRR. We then compare the Bio-RRR learning rule for $\mathbf{V}_x^\top$, which approximates the calcium plateau potential, with that of the first layer weights of this ANN. For simplicity, we focus on the RRMSE case ($s = 0$), but the interpretation of the role of the calcium plateau potential in the $\mathbf{V}_x^\top$ learning rule holds for any $s$.

The RRMSE objective given by

$$\min_{\mathbf{V}_x \in \mathbb{R}^{m \times k}} \min_{\mathbf{V}_y \in \mathbb{R}^{n \times k}} \frac{1}{T} \big\| \mathbf{Y} - \mathbf{V}_y \mathbf{V}_x^\top \mathbf{X} \big\|^2, \tag{12}$$

can be implemented as a two-layer linear ANN, where $\mathbf{V}_x^\top$ and $\mathbf{V}_y$ are the weights of the first and second layer of the network. We define $\hat{\mathbf{y}}_t = \mathbf{V}_y \mathbf{V}_x^\top \mathbf{x}_t$ as the network's prediction of the label $\mathbf{y}_t$ given input $\mathbf{x}_t$. When trained by backprop, the weight updates of this network are given by taking derivatives of the loss with respect to the weights [46]. Specifically, the learning rule for the weights of the first layer is given by:

$$\delta \mathbf{V}_x^\top \propto (\mathbf{V}_y^\top \boldsymbol{\epsilon}_t) \mathbf{x}_t^\top \quad , \quad \boldsymbol{\epsilon}_t = (\mathbf{y}_t - \hat{\mathbf{y}}_t), \tag{13}$$

where we have defined $\boldsymbol{\epsilon}_t$ as the prediction error for the sample at time $t$. The update for $\mathbf{V}_x^\top$, the weights of the first layer of the ANN, requires the computation and backpropagation of the error signal $\boldsymbol{\epsilon}_t$. A cartoon of this process is given in Fig. 2a, where the forward and backward passes are respectively denoted in blue and red. Here, the weights $\mathbf{V}_y$ are used both in the forward pass when computing the error $\boldsymbol{\epsilon}_t = \mathbf{y}_t - \mathbf{V}_y \mathbf{V}_x^\top \mathbf{x}_t$, and also their transpose in the backward pass when propagating the error back to the first layer (13). This symmetry between the forward and backward weights is a general property of SGD in deep networks but is not biologically realistic and is referred to as the "weight transport problem" [58, 59, 60]. Several solutions exist to facilitate the backpropagation of the computed error in a biologically plausible manner [61, 62, 63, 64].

Next, we show how Bio-RRR circumvents the weight transport problem. Comparing the above procedure for computing the $\mathbf{V}_x$ weight updates to that of Bio-RRR given by:

$$\delta \mathbf{V}_x^\top \propto (\mathbf{a}_t - \mathbf{Q}\mathbf{n}_t) \mathbf{x}_t^\top \tag{14}$$

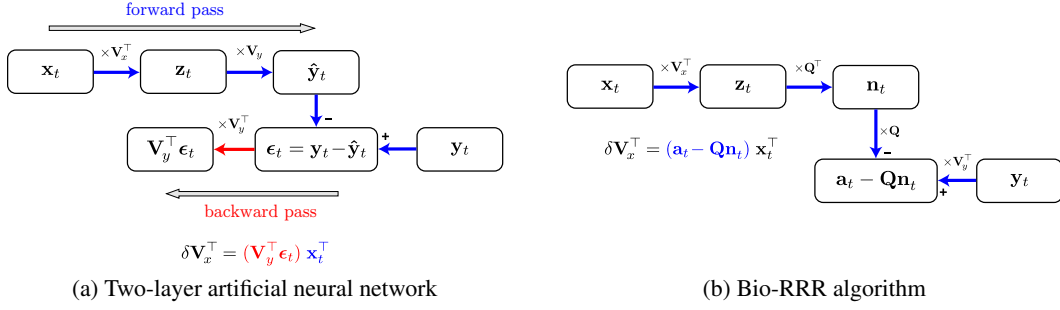

(a) Two-layer artificial neural network      (b) Bio-RRR algorithm

Figure 2: Schematic of a two-layer ANN implementation of RRR (left) and Bio-RRR (right), demonstrating the computation of the learning rule for $\mathbf{V}_x^\top$ (for $\boldsymbol{\Sigma} = \mathbf{I}_n$). The blue and red arrows respectively denote the forward and backward passes. In Bio-RRR, $\mathbf{a}_t - \mathbf{Q}\mathbf{n}_t$ (encoded in the calcium plateau potential) replaces the backpropagated error $\mathbf{V}_y^\top \boldsymbol{\epsilon}_t$ in the $\mathbf{V}_{\mathbf{x}}^\top$ learning rule.

we see that the backpropagated error term in Eq. (13) is now replaced by the term $(\mathbf{a}_t - \mathbf{Q}\mathbf{n}_t)$ which emulates the calcium plateau potential. A diagram showing how this quantity is computed is given in Fig. 2b. We see that, unlike in the backprop computation depicted in Fig. 2a, in Bio-RRR no weights are reused and therefore weight transport problem is circumvented. This is because the Bio-RRR algorithm does not require the computation of the inferred value $\hat{\mathbf{y}}_t$ and the error signal $\boldsymbol{\epsilon}_t = \mathbf{y}_t - \hat{\mathbf{y}}_t$.

Although Bio-RRR does not explicitly compute prediction error, the update for $\mathbf{V}_x^\top$ can still be interpreted in the context of error backpropagation. To this end, we look at the optimum of the objective where, from Eq. (5), we have

$$\mathbf{Q}\mathbf{Q}^\top \mathbf{V}_x^\top = \mathbf{V}_y^\top \mathbf{C}_{yx}\mathbf{C}_{xx}^{-1} \;\; \Rightarrow \;\; \mathbf{Q}\mathbf{n}_t = \mathbf{Q}\mathbf{Q}^\top\mathbf{V}_x^\top\mathbf{x}_t = \mathbf{V}_y^\top\mathbf{C}_{yx}\mathbf{C}_{xx}^{-1}\mathbf{x}_t = \mathbf{V}_y^\top\tilde{\mathbf{y}}_t,$$

where we have used $\mathbf{n}_t = \mathbf{Q}^\top \mathbf{z}_t$ and $\mathbf{z}_t = \mathbf{V}_x^\top\mathbf{x}_t$, and we have defined $\tilde{\mathbf{y}}_t := \mathbf{C}_{yx}\mathbf{C}_{xx}^{-1}\mathbf{x}_t$. As $\mathbf{C}_{yx}\mathbf{C}_{xx}^{-1} = \arg\min_{\mathbf{W}}\|\mathbf{Y} - \mathbf{W}\mathbf{X}\|_{\boldsymbol{\Sigma}}^2$ is the optimum of the rank-unconstrained regression objective, $\tilde{\mathbf{y}}_t$ is the best estimate of $\mathbf{y}_t$ given the samples received thus far. Using these quantities and the definition of $\mathbf{a}_t = \mathbf{V}_y^\top\mathbf{y}_t$, we can rewrite the quantity $\mathbf{a}_t - \mathbf{Q}\mathbf{n}_t$ and the $\mathbf{V}_x^\top$ update in Eq. (9) as

$$\mathbf{a}_t - \mathbf{Q}\mathbf{n}_t = \mathbf{V}_y^\top(\mathbf{y}_t - \tilde{\mathbf{y}}_t) \quad \Rightarrow \quad \mathbf{V}_x^\top \leftarrow \mathbf{V}_x^\top + \eta\left[\mathbf{V}_y^\top(\underbrace{\mathbf{y}_t - \tilde{\mathbf{y}}_t}_{\text{prediction error}})\right]\mathbf{x}_t^\top. \tag{15}$$

Therefore, while the error term $\mathbf{y}_t - \tilde{\mathbf{y}}_t$ and backpropagation are not present explicitly in Bio-RRR, at the optimum, the calcium plateau potential is equal to a backpropagated error signal, and the update of $\mathbf{V}_x^\top$ is proportional to the covariance of this backpropagated error signal and the input $\mathbf{x}_t^\top$.

## 7 Numerical experiments

In this section, we report the results of numerical simulations for our algorithms with $s = 0$ denoted as Bio-RRMSE and $s = 1$ denoted as Bio-CCA, and compare with current non-biologically plausible algorithms. For our experiments, we use the MediaMill dataset [65], a commonly used real-world benchmark consisting of $T = 2 \times 10^4$ samples of video data and text annotations. For our experiments, we take the predictor variables $\mathbf{X}$ to be the 100-dimensional textual features and the response variable to be the 120-dimensional visual features extracted from representative video frames.

**RRMSE.** The performance of our RRMSE algorithm on MediaMill is given in Fig. 3a in terms of the objective function in Eq. (3) with $\boldsymbol{\Sigma} = \mathbf{I}_n$. For ranks $k \in \{1, 2, 4\}$, we plot this both as a function of iteration (top) and as a function of the CPU runtime (bottom). Here, the black dashed line denotes the value of the objective at its global minimum. For comparison, we provide the performance of the iterative quadratic minimum distance (IQMD) algorithm [32] and the 2-layer ANN discussed in Sec. 6. We see that IQMD is the most sample efficient, and ANN and Bio-RRMSE are within variance of each other and match the performance of IQMD in runtime. For plots of these algorithms in the offline (batch) setting, see Sec. E.

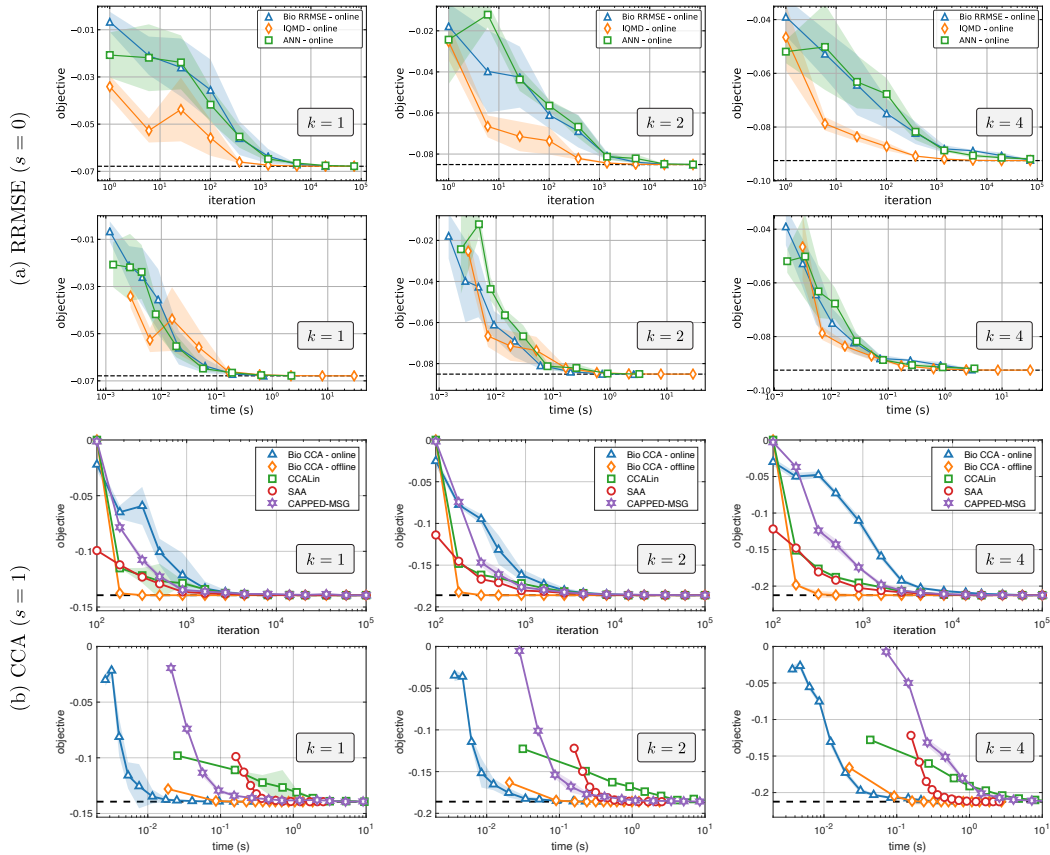

Figure 3: Comparisons of RRR algorithms for (a) RRMSE ($s = 0$) and (b) CCA ($s = 1$) in terms of the objective value Eq. (3) vs. iteration and runtime. Mean $\pm$ standard deviation over 5 runs of the experiment.

**CCA.** We evaluate the performance of our CCA algorithm on the MediaMill dataset and compare with CCALin [66] and SAA [67], which are offline algorithms, and MSG-CCA [36] which is online. In Fig. 3b, we plot the performance of the different algorithms for CCA projection dimensions $k \in \{1, 2, 4\}$ in terms of the objective function given in Eq. (3). We see that on this dataset, our offline algorithm (Bio-CCA - offline) is the most sample efficient and our online algorithm (Bio-CCA - online) is fastest in terms of CPU runtime.

For further details, including the choice of hyperparameters and plots of convergence of the RRR constraint, see supplementary materials Sec. E. For experiments comparing the performance of RRMSE and backprop on a number of standard image classification datasets, see Sec. F.

## 8 Conclusion

Employing a normative approach, we derived new offline and online algorithms for a family of optimization objectives, which include CCA and RRMSE as special cases. We implemented these algorithms in biologically plausible neural networks and discussed how they resemble recent experimentally observed plasticity rules in the hippocampus and the neocortex. We elaborated on how this algorithm circumvents the weight transport problem of backprop and how the teaching signal is encoded in a quantity that resembles the calcium plateau potential. Determining which algorithm, CCA or RRMSE, more closely resembles cortical processing would require a careful examination of synaptic plasticity in the distal compartment of pyramidal neurons.

## Broader impact

Understanding the inner workings of the brain has the potential of having a tremendous impact on society. On the one hand, this can lead to better performing machine learning algorithms and better artificial intelligent agents. On the other, understanding how the brain works can pave the way for

better treatments of psychological and neurological disorders. While this paper does not tackle these lofty broad societal goals directly, it is a small step in clarifying how information is processed in the brain.

## Acknowledgments and Disclosure of Funding

We are grateful to Jeffrey Magee and Jason Moore for insightful discussions related to this work. We further thank Nicholas Chua, Shiva Farashahi, Johannes Friedrich, Alexander Genkin, Tiberiu Tesileanu, and Charlie Windolf for providing feedback on the manuscript.

The authors did not receive any third party funding for the completion of this project.

## Footnotes

[1] Also referred to as reduced rank Wiener filter or simply reduced rank regression.

[2]There are multiple types of interneurons targeting pyramidal cells [47, 48]. The interneurons of Bio-RRR most closely resemble the somatostatin-expressing interneurons, which preferentially inhibit the apical dendrites.

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
