[Supplementary Material]

# Supplementary Materials

This is the supplementary materials section for the NeurIPS 2020 paper titled "A simple normative network approximates local non-Hebbian learning in the cortex".

## A Equivalence of CCA and RRR with $\boldsymbol{\Sigma} = \mathbf{C}_{yy}^{-1}$

In this section we show that the RRR objective in Eq. (3) is equivalent to CCA when $\boldsymbol{\Sigma} = \mathbf{C}_{yy}^{-1}$. We start with the standard CCA optimization objective

$$\max_{\mathbf{W}_x \in \mathbb{R}^{m \times k}, \mathbf{W}_y \in \mathbb{R}^{n \times k}} \mathrm{Tr}\left(\mathbf{W}_x^\top \mathbf{C}_{xy} \mathbf{W}_y\right), \quad \text{subject to } \mathbf{W}_x^\top \mathbf{C}_{xx} \mathbf{W}_x = \mathbf{W}_y^\top \mathbf{C}_{yy} \mathbf{W}_y = \mathbf{I}_k. \quad (16)$$

We then implement both constraints as Lagrange multipliers in the objective function

$$\max_{\mathbf{W}_x \in \mathbb{R}^{m \times k}, \mathbf{W}_y \in \mathbb{R}^{n \times k}} \min_{\boldsymbol{\Lambda}_x, \boldsymbol{\Lambda}_y \in \mathbb{R}^{k \times k}} \mathrm{Tr}\left[\mathbf{W}_x^\top \mathbf{C}_{xy} \mathbf{W}_y + \tfrac{1}{2}(\mathbf{W}_x^\top \mathbf{C}_{xx} \mathbf{W}_x - \mathbf{I}_k)\boldsymbol{\Lambda}_x \right.$$
$$\left. + \tfrac{1}{2}(\mathbf{W}_y^\top \mathbf{C}_{yy} \mathbf{W}_y - \mathbf{I}_k)\boldsymbol{\Lambda}_y\right], \quad (17)$$

where $\boldsymbol{\Lambda}_x$ and $\boldsymbol{\Lambda}_y$ are symmetric Lagrange multipliers. Taking derivatives with respect to $\mathbf{W}_x$ and $\mathbf{W}_y$ we find

$$\mathbf{C}_{xy} \mathbf{W}_y = \mathbf{C}_{xx} \mathbf{W}_x \boldsymbol{\Lambda}_x, \quad (18)$$
$$\mathbf{C}_{yx} \mathbf{W}_x = \mathbf{C}_{yy} \mathbf{W}_y \boldsymbol{\Lambda}_y. \quad (19)$$

Multiplying these by $\mathbf{W}_x^\top$ and $\mathbf{W}_y^\top$ respectively and using the constraints, we find $\boldsymbol{\Lambda} := \boldsymbol{\Lambda}_x = \boldsymbol{\Lambda}_y = \mathbf{W}_x^\top \mathbf{C}_{xy} \mathbf{W}_y$. Replacing $\boldsymbol{\Lambda}_x$ and $\boldsymbol{\Lambda}_y$ by $\boldsymbol{\Lambda}$ in Eqs. (18) and (19) brings us to the generalized eigenvalue problem formulation of CCA.

$$\begin{bmatrix} 0 & \mathbf{C}_{xy} \\ \mathbf{C}_{yx} & 0 \end{bmatrix} \begin{bmatrix} \mathbf{W}_x \\ \mathbf{W}_y \end{bmatrix} = \begin{bmatrix} \mathbf{C}_{xx} & 0 \\ 0 & \mathbf{C}_{yy} \end{bmatrix} \begin{bmatrix} \mathbf{W}_x \\ \mathbf{W}_y \end{bmatrix} \boldsymbol{\Lambda}. \quad (20)$$

We then solve for $\mathbf{W}_y$ in Eq. (19) to find $\mathbf{W}_y = \mathbf{C}_{yy}^{-1} \mathbf{C}_{yx} \mathbf{W}_x \boldsymbol{\Lambda}^{-1}$. Plugging this into Eq. (18) and multiplying both sides by $\mathbf{C}_{xx}^{-1}$ we arrive at

$$\mathbf{C}_{xx}^{-1} \mathbf{C}_{xy} \mathbf{C}_{yy}^{-1} \mathbf{C}_{yx} \mathbf{W}_x = \mathbf{W}_x \boldsymbol{\Lambda}^2, \quad \text{subject to } \mathbf{W}_x^\top \mathbf{C}_{xx} \mathbf{W}_x = \mathbf{I}_k.$$

Multiplying both sides by $\mathbf{W}_x^\top \mathbf{C}_{xx}$ and using the constraint we have:

$$\mathbf{W}_x^\top \mathbf{C}_{xy} \mathbf{C}_{yy}^{-1} \mathbf{C}_{yx} \mathbf{W}_x = \boldsymbol{\Lambda}^2, \quad \text{subject to } \mathbf{W}_x^\top \mathbf{C}_{xx} \mathbf{W}_x = \mathbf{I}_k.$$

The top eigenvalues of this equation can again be found via an optimization objective:

$$\min_{\mathbf{W}_x \in \mathbb{R}^{m \times k}} \mathrm{Tr}(-\mathbf{W}_x^\top \mathbf{C}_{xy} \mathbf{C}_{yy}^{-1} \mathbf{C}_{yx} \mathbf{W}_x) \quad \text{subject to} \quad \mathbf{W}_x^\top \mathbf{C}_{xx} \mathbf{W}_x = \mathbf{I}_k. \quad (21)$$

We then introduce the auxiliary variable $\mathbf{V}_y$ and rename $\mathbf{W}_x \to \mathbf{V}_x$ and arrive at:

$$\min_{\mathbf{V}_x \in \mathbb{R}^{m \times k}} \min_{\mathbf{V}_y \in \mathbb{R}^{n \times k}} \mathrm{Tr}(\mathbf{V}_y^\top \mathbf{C}_{yy} \mathbf{V}_y - 2\mathbf{V}_x^\top \mathbf{C}_{xy} \mathbf{V}_y) \quad \text{subject to} \quad \mathbf{V}_x^\top \mathbf{C}_{xx} \mathbf{V}_x = \mathbf{I}_k.$$

which is the same as Eq. (3) for $\boldsymbol{\Sigma} = \mathbf{C}_{yy}^{-1}$.

## B Naive implementation of the RRR constraint is not biologically plausible.

The RRR objective derived in Sec. 3 given by Eq. (3):

$$\min_{\mathbf{V}_x \in \mathbb{R}^{m \times k}} \min_{\mathbf{V}_y \in \mathbb{R}^{n \times k}} \mathrm{Tr}(\mathbf{V}_y^\top \boldsymbol{\Sigma}^{-1} \mathbf{V}_y - 2\mathbf{V}_x^\top \mathbf{C}_{xy} \mathbf{V}_y) \quad \text{subject to} \quad \mathbf{V}_x^\top \mathbf{C}_{xx} \mathbf{V}_x = \mathbf{I}_k.$$

includes a constraint on the weight matrices. Here, we show that if the constraint is directly implemented via a Lagrange multiplier (and not via an inequality as in Sec. 4.2), the naive neural

network implementation would not be biologically plausible. To see this explicitly, we enforce this constraint by a Lagrange multiplier $\mathbf{\Lambda}$:

$$\min_{\mathbf{V}_x \in \mathbb{R}^{m \times k}} \min_{\mathbf{V}_y \in \mathbb{R}^{n \times k}} \max_{\mathbf{\Lambda} \in \mathbb{R}^{k \times k}} \mathrm{Tr}(\mathbf{V}_y^\top \mathbf{\Sigma}^{-1} \mathbf{V}_y - 2\mathbf{V}_x^\top \mathbf{C}_{xy} \mathbf{V}_y) + \mathbf{\Lambda}(\mathbf{V}_x^\top \mathbf{C}_{xx} \mathbf{V}_x - \mathbf{I}_k).$$

If we now look at the $\mathbf{\Lambda}$ dependent synaptic update rule for $\mathbf{V}_x$ by performing gradient descent, we have:

$$\delta \mathbf{V}_x \sim \mathbf{C}_{xx} \mathbf{V}_x \mathbf{\Lambda} + \cdots. \tag{22}$$

This update includes the multiplication of two sets of synaptic weights $\mathbf{V}_x$ and $\mathbf{\Lambda}$. This would mean that the update for any component of $\mathbf{V}_x$ would require the knowledge of other components of $\mathbf{V}_x$ as well. This is not biologically plausible.

## C  Saturation of the Bio-RRR inequality constraint

Here we show that the inequality constraint imposed in Bio-RRR is saturated at its optimum in the offline setting. This was previously shown in [45]. Here we provide an alternative proof. The optimization objective is given in Eq. (4):

$$\min_{\mathbf{V}_x \in \mathbb{R}^{m \times k}} \min_{\mathbf{V}_y \in \mathbb{R}^{n \times k}} \max_{\mathbf{Q} \in \mathbb{R}^{k \times k}} \mathrm{Tr}\, \mathbf{V}_y^\top \mathbf{\Sigma}_s^{-1} \mathbf{V}_y - 2\mathbf{V}_x^\top \mathbf{C}_{xy} \mathbf{V}_y + \mathbf{Q}\mathbf{Q}^\top (\mathbf{V}_x^\top \mathbf{C}_{xx} \mathbf{V}_x - \mathbf{I}_k),$$

we first find the optimum for $\mathbf{V}_y$ by setting the $\mathbf{V}_y$ derivative to zero:

$$0 = \mathbf{V}_x^\top \mathbf{C}_{xy} - \mathbf{V}_y^\top \mathbf{\Sigma}_s^{-1} \;\Rightarrow\; \mathbf{V}_y^\top = \mathbf{V}_x^\top \mathbf{C}_{xy} \mathbf{\Sigma}_s.$$

Plugging this back into the optimization objective yields

$$\min_{\mathbf{V}_x \in \mathbb{R}^{m \times k}} \max_{\mathbf{Q} \in \mathbb{R}^{k \times k}} \mathrm{Tr}\, -\mathbf{V}_x^\top \mathbf{C}_{xy} \mathbf{\Sigma}_s \mathbf{C}_{yx} \mathbf{V}_x + \mathbf{Q}\mathbf{Q}^\top (\mathbf{V}_x^\top \mathbf{C}_{xx} \mathbf{V}_x - \mathbf{I}_k). \tag{23}$$

The equilibrium condition for this system is given by

$$0 = \mathbf{V}_x^\top \mathbf{C}_{xy} \mathbf{\Sigma}_s \mathbf{C}_{yx} - \mathbf{Q}\mathbf{Q}^\top \mathbf{V}_x^\top \mathbf{C}_{xx}, \tag{24}$$

$$0 = \mathbf{Q}^\top (\mathbf{V}_x^\top \mathbf{C}_{xx} \mathbf{V}_x - \mathbf{I}_k), \tag{25}$$

Note that Eq. (25) on its own does not imply that $\mathbf{V}_x^\top \mathbf{C}_{xx} \mathbf{V}_x = \mathbf{I}_k$. However, if we can prove that $\mathbf{Q}$ which is a $k \times k$ matrix, is full rank and has no zero eigenvalues, then Eq. (25) implies $\mathbf{V}_x^\top \mathbf{C}_{xx} \mathbf{V}_x = \mathbf{I}_k$. This is a realization of the fact that when imposing an inequality constraint, for example $f(x) > 0$, via a Lagrange multiplier $\lambda$ by optimizing $\min_x \max_{\lambda \geq 0} \lambda f(x)$, if the Lagrange multiplier at the optimum is slack $\lambda > 0$, then the inequality constraint is saturated $f(x) = 0$.

In what follows we show that at equilibrium, $\mathbf{Q}\mathbf{Q}^\top$ has no zero eigenvalues and therefore $\mathbf{Q}$ is full rank. This then proves that $\mathbf{V}_x^\top \mathbf{C}_{xx} \mathbf{V}_x = \mathbf{I}_k$ is satisfied at the optimum. To proceed, we multiply Eq. (24) by $\mathbf{V}_x$ on the right to get:

$$0 = \mathbf{V}_x^\top \mathbf{C}_{xy} \mathbf{\Sigma}_s \mathbf{C}_{yx} \mathbf{V}_x - \mathbf{Q}\mathbf{Q}^\top \mathbf{V}_x^\top \mathbf{C}_{xx} \mathbf{V}_x.$$

Plugging this back into the objective (23), we see after cancellations that the only remaining term in the objective is $-\mathbf{Q}\mathbf{Q}^\top$.

We then use Eq. (24) to solve for $\mathbf{Q}\mathbf{Q}^\top$

$$\mathbf{Q}\mathbf{Q}^\top = \tilde{\mathbf{V}}_x^\top \mathbf{C}_{xx}^{-\frac{1}{2}} \mathbf{C}_{xy} \mathbf{\Sigma}_s \mathbf{C}_{yx} \mathbf{C}_{xx}^{-\frac{1}{2}} \tilde{\mathbf{V}}_x (\tilde{\mathbf{V}}_x^\top \tilde{\mathbf{V}}_x)^{-1}, \tag{26}$$

where we have defined $\tilde{\mathbf{V}}_x := \mathbf{C}_{xx}^{\frac{1}{2}} \mathbf{V}_x$. Since $\mathbf{Q}\mathbf{Q}^\top$ is symmetric, we can take the transpose of both sides of this equation to write:

$$\mathbf{Q}\mathbf{Q}^\top = (\tilde{\mathbf{V}}_x^\top \tilde{\mathbf{V}}_x)^{-1} \tilde{\mathbf{V}}_x^\top \mathbf{C}_{xx}^{-\frac{1}{2}} \mathbf{C}_{xy} \mathbf{\Sigma}_s \mathbf{C}_{yx} \mathbf{C}_{xx}^{-\frac{1}{2}} \tilde{\mathbf{V}}_x. \tag{27}$$

Comparing Eq. (26) and Eq. (27), we see that $(\tilde{\mathbf{V}}_x^\top \tilde{\mathbf{V}}_x)^{-1}$ and $\tilde{\mathbf{V}}_x^\top \mathbf{C}_{xx}^{-\frac{1}{2}} \mathbf{C}_{xy} \mathbf{\Sigma}_s \mathbf{C}_{yx} \mathbf{C}_{xx}^{-\frac{1}{2}} \tilde{\mathbf{V}}_x$ commute. Therefore, they also commute with $(\tilde{\mathbf{V}}_x^\top \tilde{\mathbf{V}}_x)^{-1/2}$. We can use this to write $\mathbf{Q}\mathbf{Q}^\top$ as

$$\mathbf{Q}\mathbf{Q}^\top = \mathbf{U}_x^\top \mathbf{C}_{xx}^{-\frac{1}{2}} \mathbf{C}_{xy} \mathbf{\Sigma}_s \mathbf{C}_{yx} \mathbf{C}_{xx}^{-\frac{1}{2}} \mathbf{U}_x, \tag{28}$$

Figure 4: The Bio-RRR circuit with decoupled interneuron-to-pyramidal weights ($\mathbf{Q}$) and pyramidal-to-interneuron weights ($\mathbf{R}$). Following Hebbian learning rules, the weights $\mathbf{R}$ approach $\mathbf{Q}^\top$ exponentially.

where we have defined the semi-orthogonal matrix $\mathbf{U}_x^\top = (\tilde{\mathbf{V}}_x^\top \tilde{\mathbf{V}}_x)^{-\frac{1}{2}} \tilde{\mathbf{V}}_x^\top$. Plugging everything back into the objective, and remembering that the only remaining term in the objective is $-\mathbf{Q}\mathbf{Q}^\top$ we get

$$\min_{\mathbf{U}_x \in \mathbb{R}^{m \times k}} \operatorname{Tr} -\mathbf{U}_x^\top \mathbf{C}_{xx}^{-\frac{1}{2}} \mathbf{C}_{xy} \mathbf{\Sigma}_s \mathbf{C}_{yx} \mathbf{C}_{xx}^{-\frac{1}{2}} \mathbf{U}_x \text{ such that } \mathbf{U}_x^\top \mathbf{U}_x = \mathbf{I}_k. \tag{29}$$

The minimum of this objective is when $\mathbf{U}_x$ aligns with the top $k$ eigenvectors of the matrix $\mathbf{M} := \mathbf{C}_{xx}^{-\frac{1}{2}} \mathbf{C}_{xy} \mathbf{\Sigma}_s \mathbf{C}_{yx} \mathbf{C}_{xx}^{-\frac{1}{2}}$. As $\mathbf{M} = \mathbf{F}\mathbf{F}^\top$ with $\mathbf{F} := \mathbf{C}_{xx}^{-\frac{1}{2}} \mathbf{C}_{xy} \mathbf{\Sigma}_s^{1/2}$, the rank of $\mathbf{M}$ is equal to the rank of $\mathbf{F}$ which is equal to the rank of $\mathbf{C}_{xy}$. Therefore, if $\mathbf{C}_{xy}$ has at least $k$ non-zero eigenvalues, then at the optimum, $\mathbf{Q}\mathbf{Q}^\top$ has no zero eigenvalues and $\mathbf{V}_x^\top \mathbf{C}_{xx} \mathbf{V}_x = \mathbf{I}_k$ which we set out to show.

## D    Decoupling the interneuron synapses

The Bio-RRR neural circuit derived in Sec. 5, with learning rules given in Eqs. (9)−(11), requires the pyramidal-to-interneuron weight matrix ($\mathbf{Q}^\top$) to be the the transpose of the interneuron-to-pyramidal weight matrix ($\mathbf{Q}$). Naively, this is not biologically plausible and is another example of the weight transport problem discussed in Sec. 6, albeit a less severe one as both sets of neurons (pyramidal and interneurons) are roughly in the same region of the brain. Here, we show that the symmetry between these two sets of weights ($\mathbf{Q}$ and $\mathbf{Q}^\top$) follows from the operation of local learning rules.

To derive fully biologically plausible learning rules, we replace the pyramidal-to-interneuron weight matrix ($\mathbf{Q}^\top$) by a new weight matrix $\mathbf{R}$ which a priori is unrelated to $\mathbf{Q}$ (Fig. 4). We then impose the Hebbian learning rules for both sets of weights

$$\mathbf{Q} \leftarrow \mathbf{Q} + \frac{\eta}{\tau}(\mathbf{z}_t \mathbf{n}_t^\top - \mathbf{Q}) \tag{30}$$

$$\mathbf{R} \leftarrow \mathbf{R} + \frac{\eta}{\tau}(\mathbf{n_t} \mathbf{z_t}^\top - \mathbf{R}). \tag{31}$$

If we assume that $\mathbf{Q}$ and $\mathbf{R}$ assume values $\mathbf{Q}_0$ and $\mathbf{R_0}$ at time $t = 0$, after viewing $T$ samples, the difference $\mathbf{Q}^\top - \mathbf{R}$ can be written in terms of the initial values as

$$\mathbf{Q}^\top - \mathbf{R} = (1 - \eta/\tau)^T (\mathbf{Q}_0^\top - \mathbf{R_0}). \tag{32}$$

We see that the difference decays exponentially. Therefore, after viewing a finite number of samples, $\mathbf{R}$ would be approximately equal to $\mathbf{Q}^\top$ and we get back the Bio-RRR update rules.

## E    Numerical experiment details

In this section we provide further details on the numerical experiments of Sec. 7 where we validate our formalism on the MediaMill dataset [65]. As in [36], to ensure that the problem is well-conditioned, we add a small diagonal term $\varepsilon \mathbf{I}_m$ (resp. $\varepsilon \mathbf{I}_n$) to the estimates of the covariance matrices $\mathbf{C}_{xx}$ and

$\mathbf{C}_{yy}$, with $\varepsilon = 0.1$. We do this explicitly for the offline algorithms, and implicitly by adding this diagonal element to the rank one updates of the online algorithms.

Figure 3 of Sec. 7 shows performance of Bio-RRR when $s = 0$ (Bio-RRMSE) and $s = 1$ (Bio-CCA) in terms of the objective function Eq. (3):

$$\min_{\mathbf{V}_x \in \mathbb{R}^{m \times k}} \min_{\mathbf{V}_y \in \mathbb{R}^{n \times k}} \mathrm{Tr}(\mathbf{V}_y^\top \mathbf{\Sigma}_s^{-1} \mathbf{V}_y - 2\mathbf{V}_x^\top \mathbf{C}_{xy} \mathbf{V}_y) \quad \text{subject to} \quad \mathbf{V}_x^\top \mathbf{C}_{xx} \mathbf{V}_x = \mathbf{I}_k.$$

Since this objective has a whitening constraint which is not necessarily enforced in other algorithms we compare with, when measuring the performance of each algorithm, we manually enforce this constraint at each time step. Similarly, the weight $\mathbf{V}_y$ is not present in the same form in all algorithms, we therefore integrate it out in the objective, placing it at its optimum $\mathbf{V}_y = \mathbf{\Sigma}_s \mathbf{C}_{yx} \mathbf{V}_x$. Explicitly, we plot the value of the quantity

$$-\tilde{\mathbf{V}}_x^\top \mathbf{C}_{xy} \mathbf{\Sigma}_s \mathbf{C}_{yx} \tilde{\mathbf{V}}_x \quad \text{where} \quad \tilde{\mathbf{V}}_x = (\mathbf{V}_x^\top \mathbf{C}_{xx} \mathbf{V}_x)^{-1/2} \mathbf{V}_x. \tag{33}$$

By explicitly imposing the whitening constraint and integrating $\mathbf{V}_y$ out, this quantity has the advantage of measuring only the correct alignment of the latent space $\mathbf{Z} = \mathbf{V}_x \mathbf{X}$ and not the overall magnitude. This makes for a fair comparison, especially when considering methods such as IQMD [32] and the 2-layer ANN of Sec. 6, which do not impose any constraints on the overall magnitude of the latent space.

In our experiments, we run the offline algorithms for $2 \times 10^4$ iterations (equal to one epoch) and the online algorithms for $10^5$ iterations (5 epochs). For each algorithm, we run the experiment 5 times with random initializations and random sample order in the online case and report the mean $\pm$ standard deviation of the quantity in Eq. (33).

To directly verify that the Bio-RRR algorithm indeed satisfies the whitening constraint as claimed in Sec. 4, we plot the deviation of the variables from the constraint at each time point. Explicitly, Fig. 5 shows the value of the quantity $\|\mathbf{V}_x^\top \mathbf{C}_{xx} \mathbf{V}_x - \mathbf{I}_k\|^2/k$ on the MediaMill dataset for both RRMSE ($s = 0$) and CCA ($s = 1$) in the online setting. We see that, at convergence, the RRR whiteness constraint is indeed satisfied.

Figure 5: The deviation of the RRMSE solution (left) and CCA solution (right) from orthonormality constraint in terms of $\|\mathbf{V}_x^\top \mathbf{C}_{xx} \mathbf{V}_x - \mathbf{I}_k\|^2/k$ in the online setting. Mean $\pm$ standard deviation over 5 runs of the experiment.

In the following, we provide further details in the individual RRMSE and CCA experiments.

**RRMSE.** The RRMSE experiments are run in Python on a 2019 MacBook Pro 13" with 2.8GHz quad-core 8th-generation Intel Core i7 (i7-8569U CPU at 2.80GHz) processor. Of the three methods compared, IQMD does not have any hyperparameters. For ANN and Bio-RRMSE, which include learning rates as hyperparameters, we parametrize each individual learning rate as $\eta = \frac{\eta_0}{1 + t/N}$ where $\eta_0$ encodes the learning rate at the start of training and $N$ encodes the rate of decay of the learning rate. Furthermore, as the plasticity rate of different neurons are not necessarily the same, for increased realism, we allow for unequal learning rates for the different weights of both Bio-RRMSE and ANN. For each algorithm and each value of $k$, we perform a coarse grid search covering two decades for each parameter, starting with the largest value for which the algorithm does not diverge. We find that the performance of neither algorithm is very sensitive to the choice of $N$ and $\eta_0$. In the online setting

(with results shown in Fig. 3a), for Bio-RRMSE we use $\eta_x = \frac{1.5}{1+t/500}$, $\frac{3.5}{1+t/200}$, and $\frac{3}{1+t/7000}$ for $k = 1, 2, 4$ with $\eta_y = \eta_q = 0.002 \times \eta_x$ in each case. Here $\eta_x$, $\eta_y$ and $\eta_q$ are respectively the learning rate for the $\mathbf{V}_x$, $\mathbf{V}_y$ and $\mathbf{Q}$ synaptic weight matrices. For ANN, $\eta_x = \frac{0.5}{1+t/500}$ and $\eta_y = 0.5 \times \eta_x$ for $k = 1, 2, 4$.

Figure 6: Comparisons of RRMSE algorithms in the offline setting in terms of the objective value Eq. (33) vs. iteration and runtime. Mean $\pm$ standard deviation over 5 runs of the experiment.

The performance of the RRMSE algorithms in the offline setting in terms of the quantity in Eq. (33) (with $\mathbf{\Sigma} = \mathbf{I}_n$) is provided in Fig 6. We see again the IQMD is the more efficient algorithm and ANN and Bio-RRMSE have comparable performance in terms of sample efficiency. However, in this case, Bio-RRMSE is faster than ANN in terms of CPU runtime. In these experiments, for Bio-RRMSE we use $\eta_x = \frac{25}{1+t/500}$, 24, and $\eta_x = 20$, for $k = 1, 2, 4$ again with $\eta_y = \eta_q = 0.002 \times \eta_x$ in each case. For ANN we use $\eta_x = \eta_y = \frac{1}{1+t/20000}$ for $k = 1$, $\eta_x = \eta_y = 1$ for $k = 2$, $\eta_x = \eta_y = 0.8$ for $k = 4$.

**CCA.** The CCA experiments are run in Matlab on a Windows PC with an Intel Core i7-4770k processor clocked at 4.2Ghz. The performance of Bio-CCA as well as competing algorithms in both online and offline setting, in terms of the quantity in Eq. (33) (with $\mathbf{\Sigma} = \mathbf{C}_{yy}^{-1}$), is shown in Fig. 3b of Sec. 7. In this case, because of the $\mathbf{C}_{yy}$ factors in the objective function (2), a simple two-layer artificial neural network implementation is not possible. In this experiment the state-of-the-art competitor to Bio-CCA in the online setting is Capped-MSG [36] for which we use $K_{\text{cap}} = 6k$ and $\eta_t = \frac{0.1}{\sqrt{t-100+1}}$. For Bio-CCA, in the online setting, we use $\eta_x = \frac{3}{1+t/100}$, $\frac{2.5}{1+t/100}$, $\frac{1.2}{1+t/1000}$ for $k = 1, 2, 4$, and in the offline setting we use $\eta_x = 10, 10, 8$ for $k = 1, 2, 4$. In all cases we use $\eta_y = \eta_q = 0.02 \times \eta_x$.

## F  More numerical experiments

For a more detailed comparison of Bio-RRMSE and the backprop-trained ANN discussed in Sec. 6, we looked at a number of image classification datasets (MNIST [68], Fashion MNIST [69], CIFAR-10, and CIFAR-100 [70]). In all these cases, we take $\mathbf{X}$ to be the vectorized sample images in pixel space and take $\mathbf{Y}$ to be the one-hot vector of image labels. Figure 7 shows the results of this experiment in terms of the objective function given in Eq. (33) for one rank per dataset ($k = 1, 2, 4, 8$ respectively for MNIST, FMNIST, CIFAR-10, CIFAR-100). In all cases, the performance of Bio-RRMSE is comparable to the performance of backprop. The hyperparameters chosen for these experiments are given in Tab. 1.

Figure 7: Comparison of RRMSE vs backprop for a number of image classification datasets in terms of the objective value in Eq. (33) with $s = 0$.

|  | Bio-RRMSE | | | Backprop | |
|---|---|---|---|---|---|
|  | $\eta_x$ | $\eta_y$ | $\eta_q$ | $\eta_x$ | $\eta_y$ |
| MNIST | $\frac{0.01}{1+t/10^3}$ | $\frac{0.01}{1+t/10^3}$ | $\frac{0.003}{1+t/10^3}$ | $\frac{0.02}{1+t/10^3}$ | $\frac{0.02}{1+t/10^3}$ |
| FMNIST | $\frac{0.013}{1+t/10^3}$ | $\frac{0.013}{1+t/10^3}$ | $\frac{0.005}{1+t/10^3}$ | $\frac{0.018}{1+t/10^3}$ | $\frac{0.018}{1+t/10^3}$ |
| CIFAR-10 | $\frac{0.01}{1+t/1.5\times10^4}$ | $\frac{0.002}{1+t/1.5\times10^4}$ | $\frac{0.002}{1+t/1.5\times10^4}$ | $\frac{0.0065}{1+t/10^4}$ | $\frac{0.0065}{1+t/10^4}$ |
| CIFAR-100 | $\frac{0.025}{1+t/4\times10^4}$ | $\frac{0.001}{1+t/4\times10^4}$ | $\frac{0.002}{1+t/4\times10^4}$ | $\frac{0.0065}{1+t/1.1\times10^4}$ | $\frac{0.0065}{1+t/1.1\times10^4}$ |

Table 1: Hyperparameter choices for the linear experiment with results reported in Fig. 7.