[Reviews · NeurIPS 2020]

Review 1

Summary and Contributions: The authors derive online versions of RRMSE and CCA, and map it to pyramidal neurons with distal/proximal inputs, inhibition and Ca^{2+} plateau potentials. They show empirical results that these local online rules work well.

Strengths: The authors derive simple models of the online variants of RRMSE and CCA, which is new to my knowledge. The use of such local learning rules might lead to efficient implementations, say on neuromorphic hardware.

Weaknesses: The empirical evaluation is one of the weakest aspects of the paper. The fact that this is done on only one, seemingly arbitrarily chosen, dataset diminishes the significance of the results. I would have liked to see evaluation on more standard datasets. There are some aspects of the biological mapping that may not be biologically plausible: - Only linear model are considered. In biology, pyramidal cells are known to have many non-linear effects. - Inhibition is considered with tied input and output weights. Need more details on which type of inhibitory neurons ave this property. - Distal plasticity ignores known data from (Milstein et al. 2020; Magee and Grienberger 2020 [1]) the first of which is cited in the paper for other reasons. - The authors don't consider dendritic spikes and calcium plateau potentials as events. - V_y updates use activity of soma and input to distal compartments, which is quite unusual for biological rules. - Ca^{2+} plateau potential are generally know to affect distal dendrites, which is not considered here. - They don't consider binary or spiking data which could be more biologically plausible. An analysis of what happens when moving from full dataset to single samples would have been useful. The authors consider algorithms that are used for dimensionality reduction (as mentioned in their discussion), but need to specify which pyramidal cells in the brain are known to have this property. A discussion of how representations learnt using RRMSE or CCA can be used by further processing layers can strenghten the paper. [1] Magee, J.C., and Grienberger, C. (2020). Synaptic Plasticity Forms and Functions. Annual Review of Neuroscience 43.

Correctness: Some of the decisions taken when mapping to biology ignores or stretches known data. The empirical evaluation is done on only one, seemingly arbitrarily chosen, dataset.

Clarity: The paper is generally clearly written, and the authors motivate the problem quite nicely.

Relation to Prior Work: (Milstein et al. 2020; Magee and Grienberger 2020) discuss potential plasticity mechanisms where Ca^{2+} plays a role, which is not taken into consideration. (See list of biologically implausible aspects of the model.) Other work on role of pyramidal neurons and plasticity there [2,3,4] could be discussed. [2] Larkum, M. (2013). A cellular mechanism for cortical associations: an organizing principle for the cerebral cortex. Trends in Neurosciences 36, 141–151. [3] Kampa, B.M., Letzkus, J.J., and Stuart, G.J. (2007). Dendritic mechanisms controlling spike-timing-dependent synaptic plasticity. Trends in Neurosciences 30, 456–463. [4] Gidon, A., Zolnik, T.A., Fidzinski, P., Bolduan, F., Papoutsi, A., Poirazi, P., Holtkamp, M., Vida, I., and Larkum, M.E. (2020). Dendritic action potentials and computation in human layer 2/3 cortical neurons. Science 367, 83–87.

Reproducibility: Yes

Additional Feedback: Update post-author-response: I am satisfied with the new empirical results. The author also promise to make it very clear which biological aspects are supported by currently known data and which is speculation. I am revising my score to accept. Minor: On l.121, the authors probably mean eqn. (3) not (18) Is the MediaMill dataset conducive for linear models? What happens on a dataset that is not, with these learning rules?


Review 2

Summary and Contributions: RRR and CCA are two classical statistical techniques with well-known solutions. In this paper, the authors design new online learning algorithms for RRR and CCA, such that learning works under biological constraints, or, specifically, such that all learning is local. The results are given a biological interpretation and are demonstrated in numerical experiments.

Strengths: Overall, I found the paper interesting. In principle, RRR (and CCA) could be implemented by two-layer linear networks, and then learnt via back-propagation, for which, by now, several candidates for local learning rules exist. However, the authors here choose a different path, which I think is interesting in its own right. Essentially, they eliminate the middle layer, design a novel online algorithm and then go to great length to give their algorithm a biophysical interpretation. I think this new way of thinking about what essentially is a two-layer network provides an interesting perspective on a classical learning problem.

Weaknesses: The biological interpretation is maybe the weakest part of the paper, as it is quite speculative, but the authors do not clearly state that. Basically, I find the online algorithm intriguing, and I find that the biological interpretation consists of mixture of interesting ideas and some claims that are plainly wrong (see below).

Correctness: The neural circuit implementation suffers from the fact that it's a loose interpretation of the terms in equations (9)-(11). These equations are, at most, those of a rate network, and there is no established mathematical framework to consider compartments, plateau-potentials, etc. in rate networks. Therefore, many aspects of the biological circuit implementation hang in the air, and seem speculative at best. Some also seem simply wrong. (a) For instance, the output of the pyramidal cell is assumed to be z=Vx. That pre-supposes that inputs from the apical dendrite (a-Qn) either cancel out to zero, or never make it into the soma. Either seems wrong, especially if those inputs are supposed to give rise to plateau potentials which will almost certainly cause rapid firing of the output. (b) The term a-Qn is supposed to be a 'calcium plateau potential traveling down the apical shaft.' However, Calcium plateau potentials are really like slow calcium spikes, meaning that, to first order, they are stereotypical events. They are not good candidates to carry graded signals. (c) Also, you use pyramidal cells and interneurons, but they are not excitatory or inhibitory neurons (as both cell types can excite or inhibit their downstream partners). Please clarify. More generally, I dont mind (even wild) speculation about matches to biology, but it should be made crystal clear what's speculation, what's a hypothesis, and what's fact. The paper does not generally succeed in doing so.

Clarity: The technical parts are generally well written. The one section I struggled with was the link to backprop (Figure 2), which is quite intriguing, but I found the text and logic hard to follow. I think it would be of interest to the field if the authors could expand on these ideas (as many people are interested in biologically plausible backprop)

Relation to Prior Work: Yes.

Reproducibility: Yes

Additional Feedback:


Review 3

Summary and Contributions: This paper finds a local, online algorithm for performing reduced rank regression and canonical correlation analysis, and also any interpolation of the two. Update post-author-response: thanks for your reply, but linearity was not a big issue for me.

Strengths: The approach is elegant, and the circuit matches what's seen in hippocampus and neocortex -- including plateau potentials.

Weaknesses: No major weaknesses. A very minor weakness is that they're essentially doing linear dimensionality reduction, but it's likely that's a computation that the brain does.

Correctness: I think so.

Clarity: Yes. A bit dense, but that's because of the subject matter.

Relation to Prior Work: I'm not an expert, so I can't really say.

Reproducibility: Yes

Additional Feedback: None.


Review 4

Summary and Contributions: ***** UPDATE ***** I appreciate the new empirical results and am increasing my score by 1 point as a result. My remaining hesitations center on whether the paper effectively balances (1) model simplicity, (2) ability to account for experimental findings and (3) a computationally powerful algorithm. I was hoping for more on (1) and (2) in part because RRR is a relatively computationally weak/simple algorithm - maybe on par with PCA, which can be accomplished with a much simpler model like Oja's Rule. There have also already been published a number of multi-compartmental neuron models that implement interesting algorithms while explaining some subset of experimental findings (and I suspect that many more are possible), and it's not clear to me that this model is more parsimonious or explanatory than existing ones. If accepted, I would be happy to see some of the above briefly addressed in the paper. ***** The authors develop offline and online learning rules defined to achieve a Reduced Rank Regression objective. By mapping the online rule onto various intra-neuron signals in a neural model with segregated proximal and distal compartments, they develop the rule into a model of local cortical plasticity driven by supervisory signals. They further present the experimental evidence in support of the neural dynamics predicted by the online rule, and compare the performance of the offline and online rules to several baselines on the MediaMill benchmark dataset.

Strengths: The paper is well-written and theoretically well-developed. Since the authors start with a family of well-defined and well-known objectives in RRR and work backwards to the learning rules, the results are less dependent on simulations. Finally, the paper does an admirable job of reviewing the experimental evidence and discussing potential issues with the model in the final section.

Weaknesses: The normative approach is appealing, but the model lacks any form of temporal dynamics which is limiting as a biologically plausible model of neural dynamics or plasticity. The linear approach is also limited both as a description of neurons and in computational capacity. These limitations are challenging in comparison to existing models of learning dynamics built on segregated neuron models with richer dynamics (e.g. Guergiuev as cited, or Urbanczik and Senn 2014 below). I would like to see a more convincing argument in the Introduction for the significance of this new model over the existing models. While the authors nicely show how the learning rule can be interpreted in terms of prediction error "at optimum", I would like to know whether the interpretation approximately holds more generally (e.g. during learning); potentially in the supplemental.

Correctness: The claims, method and methodology appear to be correct.

Clarity: Generally the paper is well-written. A few more sentences describing the baseline algorithms in Section 6 would be appreciated, if it can fit.

Relation to Prior Work: As mentioned above, I think that the paper would benefit from a clearer argument for the value of this model over similar existing learning rules. It's not clear to me that the existing models "optimize only the synaptic weights while postulating the architecture and activity dynamics" more than this work. To the list of existing research, I would add Urbanczik and Senn, "Learning by the Dendritic Prediction of Somatic Spiking", which similarly develops a compartmental learning rule with teaching signals.

Reproducibility: Yes

Additional Feedback:

[Author Response · NeurIPS 2020]

We thank the reviewers for their thoughtful feedback. We are encouraged that they found our approach interesting and
elegant [**R2**, **R3**], and our algorithm novel and theoretically well developed [**R1**, **R4**]. We are pleased that **R2** recognized
the interest in our approach in the backprop community, and as a new perspective on a classical learning problem that is
commonly solved using backprop.

**Recap:** The goal of our work is to establish a bridge between specific computational tasks and experimentally observed
biological phenomena. To this end, we formalize the observation that pyramidal neurons combine instructive and sensory
inputs in an optimization problem reflecting the simplest possible computational task. By solving this optimization
problem in the online setting we derive a neural network with local learning rules. Interestingly, this simple linear
model captures essential aspects of cortical microcircuits including the connectivity structure and the non-Hebbian
nature of the learning rules in pyramidal neurons. Furthermore, this approach lets us interrogate which aspects of a
detailed model are essential for performing this optimization task. To better highlight the goal of this work, we will
change the title to "A simple normative network approximates local non-Hebbian learning in the cortex."

Below we answer some specific comments, but will incorporate all feedback in the final version.

[**R1**, **R2**, **R4**] **Relationship to biology and to prior work.** The price paid for the clarity of the normative approach is
that it does not reproduce every known biological observation. Our results highlight which experimental observations
about physiology are important for the circuit to implement this supervisory algorithm. We are grateful to the reviewers
for pointing out relevant references [Kampa et al 2007, Urbanczik and Senn 2014, Gidon et al 2020]. We will add a
"Related works" section where we will cite and discuss these, as well as [Sacramento et al 2018] in detail and point out
the differences and the similarities. We will also amend the experimental evidence section of Sec. 4 to clarify what is
speculation, what is hypothesis, and what is fact. We will also clearly delineate realistic and unrealistic features of the
model including the lack of apical contribution to the output of the neuron, the slow and binary nature of $Ca^{2+}$ plateaus
in rodents, the lack of excitatory/inhibitory distinction among linear neurons.

[**R1**] **The empirical evaluation is one of the weakest aspects of the paper.** We reran the numerical experiments of
Sec. 6 on five standard datasets of varying difficulties: MNIST, Fashion MNIST, CIFAR10, CIFAR100 and XRMB
JW11 (a datset of acoustic and articulation measurements commonly used for testing algorithms for CCA and RRMSE)
for ranks $k = 1, 2, 4, 8, 16$. Figure 1 shows a fraction of these results (one rank per dataset and only showing comparison
with backprop for clarity and space constraints). In all cases, the performance of Bio-RRR, measured by the objective
value in Eq. (3), is comparable to an ANN performing the same task but trained with backprop (as described in Sec. 5).
We will include the full results in the supplementary materials.

Figure 1: Comparison of the empirical performance of Bio-RRR vs. backprop on an artificial neural network. The
dashed black line denote the value of the objective at its global minimum. The figures follow the same conventions as
in Fig. 3 of the manuscript.

29
[**R1**] $Ca^{2+}$ **plateau potentials are generally known to affect distal dendrites.** As we understand it, experimental
observations regarding plasticity of distal dendrites range from those giving more weight to $Ca^{2+}$ plateaus (in the
hippocampus [Golding et al 2002]), to those giving more weight to the back-propagating action potentials (in the
neocortex [Sjöström et al 2006]). In our learning rules for distal synapses Eq. (10), the relative significance of the
terms $\mathbf{a}_t \mathbf{y}_t$ and $\mathbf{z}_t \mathbf{y}_t$ to learning is determined by the parameter $s$. The solvability of our normative model allows us to
analytically explore the task performed by the circuit as we vary the distal learning rules. Explicitly, we can verify that
the two extremes of $s = 0$ and $s = 1$ correspond to the statistical tasks RRMSE and CCA.

[**R1**, **R3**, **R4**] **Linearity of model is unrealistic and computationally limiting.** As mentioned above, the linearity of
the model is a price paid for achieving a simple and analytically tractable model. Because of this, we are limited to
considering models that perform dimensionality reduction. However, models of dimensionality reduction are known
to be computationally effective and useful for learning. For example, CCA is intimately related to the information
bottleneck problem and approaches which find features with the highest amount of mutual information [Chechik et al.
2005]. Previous experience shows that nonlinear extensions of such well understood linear models add useful features
like dimension expansion, while retaining many aspects of network structure and of learning.

[Meta-Review · NeurIPS 2020]

The reviewers agreed that this was an interesting model that provides interesting algorithmic ideas for biological theories. The rebuttals addressed the most pressing concerns, and so it was agreed that the paper should be accepted.